

# New insights of the influence of ocean circulation on the sedimentary distribution in the Southwestern Atlantic margin (23ºS to 55ºS) based on Nd and Pb isotopic fingerprinting

5  Michel Michaelovitch de Mahiques[1,2], , Roberto Violante[3], Paula Franco-Fraguas[4], Leticia Burone[4], Cesar Barbedo Rocha[5,6], Leonardo Ortega[7], Rosangela Felicio dos Santos[1], Bianca Sung Mi Kim[1], Rubens Cesar Lopes Figueira[1], Marcia Caruso Bícego[1]

1 Oceanographic Institute of the University of São Paulo. 05508-120, São Paulo, Brazil

10  2 Institute of Energy and Environment of the University of São Paulo. 05508-010, São Paulo, Brazil

3 Servicio de Hidrografía Naval. C1270ABV, Buenos Aires, Argentina

4 Facultad de Ciencias, Universidad de La Republica. 11400, Montevideo, Uruguay

5 Woods Hole Oceanographic Institution. 02543-1050, Woods Hole, USA

Presently at the Department of Marine Sciences, University of Connecticut. 06340, Groton, USA

7 Departamento de Biología Pesquera, Dirección Nacional de Recursos Acuáticos. 11200 Montevideo, Uruguay

Correspondence to: Michel Michaelovitch de Mahiques (mahiques@usp.br)

**Abstract.** In this work, we provide an extensive inventory of Pb and Nd radiogenic isotopes in surface sediments from the
Southwestern Atlantic margin, aiming to interpret the role played by ocean circulation in sediment distribution.  There are latitudinal trends for Pb and Nd isotopes, reflecting the different current systems acting on the margin.  The utilization of sediment fingerprinting allowed us to associate the isotopic signatures to the main oceanographic forcings in the area.  We recognized differences between the Nd and Pb sources for the sediments to the Argentinean shelf, carried by the Subantarctic Shelf Water, and slope, transported by deeper flows. Sediments from Antarctica extend up to the Uruguayan margin, carried
by the Upper- and Lower Circumpolar Deep Water.  Our data confirm that, for shelf and intermediate (up to 1,200 m water depth) areas, the transfer of sediments from the Argentinean margin to the North of 35ºS is limited by the Subtropical Shelf Front and the recirculated Antarctic Intermediate Water.

On the southern Brazilian margin, it is possible to recognize the northward influence of the Río de la Plata sediments carried by the Plata Plume Water. This influence is limited by the southward flow of waters transported by the Brazil Current. Finally,
we propose that the Subtropical Shelf Front and the Santos Bifurcation act as boundaries of geochemical provinces in the area. Finally, a qualitative model of sediment sources and transport is provided for the Southwestern Atlantic margin.

# 1 Introduction

Long half-life radiogenic elements, such as Sr, Pb, and Nd, are efficient tools in the study of sediment transport in continental margins.  Weldeab et al. (2002) utilized Sr and Nd isotopes to trace the main pathways of riverine suspended sediments on the





Eastern Mediterranean. Kessarkar et al. (2003) used the same isotopes, together with clay mineralogy, to evaluate the contribution of the Ganges-Brahmaputra river sediments to the western Indian margin.

Roy et al. (2007) used εNd values together with $^{40}Ar/^{39}Ar$ ages from sediment samples around Antarctica to determine the primary sources of sediments to the Southern Ocean. More recently, Maccali et al. (2018) analyzed the flow of ice-transported sediments from the Arctic Ocean using Nd, Pb, and Sr isotopes. Subha Anand et al. (2019) conducted an extensive analysis

of sediment provenance and transport in the Indian Ocean, combining Rare Earths and Sr and Nd isotopes to analyze potential sources of sediments and sediment pathways in the area.

A recent synthesis of potential sources and water masses transport in the South Atlantic was provided by Beny et al. (2020). In that study, the authors combined grain-size, clay mineralogy, and Nd, Pb, and Sr isotopes to propose a deep-water mass evolution for the last ca. 30,000 years in the region. de Mahiques et al. (2008) provided the first regional characterization of

Nd and Pb isotope signatures of the upper margin surface sediments in the Southwest Atlantic. They proposed a qualitative model for sediment sources, recognizing the Andes, the Río de la Plata drainage system, and the Pre-Cambrian southern Brazilian rocks, as the primary sources of the sediments to the Argentinean and Uruguayan margin, the southern and the southeastern Brazilian margins, respectively. That pioneering work filled an extensive area without information about εNd signatures (Jeandel et al., 2007; Blanchet, 2019). Another approach for sediment sources and pathways in the Southwestern

Atlantic margin was provided by Razik et al. (2015). These authors argued for a mixed Rio de la Plata - Andean origin for the sediments of the upper slope off southern Brazil.

In this paper, we use the concept of sediment fingerprinting to interpret the sediment transport on the Southwestern Atlantic margin, using Nd and Pb isotopes. In a focused area, the isotope values are compared with outputs of current modeling to understand the role of oceanographic boundaries in the distribution of sediments along the area.

## 55  2 Study Area

The study area comprises a southwestern Atlantic margin sector, from the parallels 23º00'S to 54º10'S, corresponding to a linear extension of about 3,500 km (Figure 1). Syntheses of the main geological and oceanographic processes can be found in Hernandez-Molina et al. (2009), Franco-Fraguas et al. (2014), Nagai et al. (2014a), Nagai et al. (2014b), Violante et al. (2014), Hernández-Molina et al. (2015), Franco-Fraguas et al. (2016), Violante et al. (2017a), Burone et al. (2018), Piola et al. (2018),

and Piola and Matano (2019), among several others.



**Figure 1. Location of the study area, displaying the main flows, oceanographic boundaries, and sampling stations. Violet thick line:**
**Malvinas Current (MC); thin blue line: Subantarctic Shelf Water (SASW); thin brown line: Río de la Plata Plume (RdlPP); thick**
**red line: Brazil Current (BC); shaded rectangles: Subtropical Shelf Front (STSF) and Brazil – Malvinas Confluence (BMC);**
**dashed red line Santos Bifurcation (SB) and Intermediate Western Boundary Current (IWBC). Other Abbreviations: Antarctic**
**Intermediate Water (AAIW), Upper Central Deep Water (UCDW), North Atlantic Deep Water (NADW), and Lower Central Deep**
**Water (LCDW). Symbols: Argentina (●), Punta del Este Basin (□), Río de la Plata (✕), Pelotas Basin (△), and Santos Basin (☆).**
**Bottom scale: topography in meters**



## 2.1 Morphology

The Southwestern Atlantic margin is a typical segmented volcanic-rifted margin, where several transverse basins are
recognized (Bassetto et al., 2000; Moulin et al., 2010; Soto et al., 2011). Its origin and evolution are intrinsically related to
the opening of the South Atlantic (Nürnberg and Müller, 1991), whose rifting processes first occurred during the Triassic
(Lovecchio et al., 2020) but effectively took place in the Jurassic and Cretaceous.

There is a general trend of narrowing the margin's width towards the North (Urien and Ewing, 1974; Zembruscki, 1979; Parker
et al., 1996; Violante et al., 2017a). The shelf width varies from 850 km to the south to 70 km in its northernmost limit; the
shelf-break depth ranges from 80 m, in southern Brazil, to 200 meters, in Uruguay (Zembruscki, 1979; Muñoz et al., 2010;
Lantzsch et al., 2014). As a rule, the shelf morphology is relatively flat, but sequences of scarps and terraces are recognized
along the whole continental shelf at varying water depths (Corrêa, 1996; Parker et al., 1996; Baptista and Conti, 2009). The
sequence of terraces and adjacent scarps is associated with sea-level stabilization intervals after the Last Glacial Maximum
(Corrêa, 1996; Violante et al., 2014). Relicts of complex barrier islands, sandbanks, ancient beaches, and shorelines, developed
during the post-Last Glacial transgression, are also found along the shelf (Urien and Ewing, 1974; Urien et al., 1980a; Parker
et al., 2008; Violante et al., 2014; Cooper et al., 2018).

In the Uruguayan inner shelf, an elongated depression, representing the Río de la Plata (RdlP) paleo-valley, extends in the SW-
NE direction along the Uruguayan coast (Urien et al., 1980a; López-Laborde, 1999; Cavallotto et al., 2005; Lantzsch et al.,
2014) and reaching the southern Brazilian shelf (Corrêa et al., 2014). This depression was about 35-km wide and up to 50-m
deep before it became partly filled by sediments since deglacial times (Lantzsch et al., 2014).

The continental slope presents a high variable morphology, including contouritic terraces, channels, mounds, erosive surfaces,
and sediment drifts all along the area (Duarte and Viana, 2007; Hernández-Molina et al., 2010; Preu et al., 2013; Hernández-
Molina et al., 2015) and canyons (Voigt et al., 2013; Bozzano et al., 2017; Franco-Fraguas et al., 2017; Violante et al., 2017b;
Warratz et al., 2019). Along the margin, the contouritic features and submarine canyons actively interact so that mixed
contouritic-gravitational erosive and depositional features are common.

Mega-slides (Reis et al., 2016; Franco-Fraguas et al., 2017) and carbonate mounds (Carranza et al., 2012; Maly et al., 2019;
Steinmann et al., 2020) are also present along the margin. Another impressive feature of this margin is the Rio Grande Cone.
This fan-shaped feature extends from the outer shelf to the base of the slope, representing the deposition of more than 10 km
of terrigenous sediments (Miller et al., 2015a; Alberoni et al., 2019). Some studies associate this feature with the input of
sediments from the Río de la Plata under conditions of lowstand (Corrêa et al., 2014; Razik et al., 2015).

The base of the slope can extend up to 4500 m (Violante et al., 2017b), but is significantly shallower in the North, where an
intense halokinesis promoted the development of a unique geomorphological feature, the São Paulo Plateau (Kumar and
Gambôa, 1979; Mohriak et al., 2008; de Almeida and Kowsmann, 2016).



## 2.2 Sedimentary cover

The Southwestern Atlantic margin is dominated by a terrigenous, siliciclastic sedimentary cover, with extensive sand sheets (Lonardi and Ewing, 1971; Frenz et al., 2003; Figueiredo and Madureira, 2004). The Argentinean and Uruguayan shelves are capped mainly by a 5 to 15 m thick post-Late Glacial transgressive sandy sheet (in general decreasing thickness towards the south) composed of dominant medium to fine sands (sometimes muddy), with varying amounts of shells (more abundant in the Uruguayan shelf) and gravels (more abundant in the Patagonian shelf).

Sandy and shelly sediments are mainly relicts of coastal and inner shelf environments evolved during Pleistocene transgressive-regressive events (Kowsmann and Costa, 1979; Urien et al., 1980b; Lantzsch et al., 2014). Therefore, they are considered relict and palimpsest, whereas gravelly-dominated sediments on the southern Argentinean shelf result from glacifluvial origin. More recent works emphasize the existence of mud depocenters as potential fates of modern sediments on the southern Brazilian shelf (Nagai et al., 2014a; de Mahiques et al., 2017; Lourenço et al., 2017; de Mahiques et al., 2020).

In the slope and rise, there is a prevalence of very fine sands and silty sands, resulting from exclusively submarine processes occurring across- (gravitational) and along- (contouritic) slope, together with pelagic sedimentation (Violante et al., 2010; Bozzano et al., 2011; Franco-Fraguas et al., 2016; Schattner et al., 2020). However, coarse sands and gravels occur at or near the head of submarine canyons and in contouritic channels and moats (Lonardi and Ewing, 1971; Bozzano et al., 2011; Reis et al., 2016; Franco-Fraguas et al., 2017). On the slope off southern Brazil, Razik et al. (2015) indicate increasing grain size

towards coarse sands due to sediment remobilization and redistribution due to upwelling and downwelling resulting from eddies and vertical water movement generated by the meandering Brazil Current.

Concerning mineralogy, there is a consensus about three primary continental sources for the terrigenous sediments capping the Southwestern Atlantic margin: Andean, Paraná-Plata basin, and Brazilian craton (Teruggi, 1954; Etchichury and Remiro, 1960, 1963; Berkowsky, 1978; Potter, 1984; Berkowsky, 1986; Campos et al., 2008; Nagai et al., 2014a).

Andean-sourced sediments, defined by Teruggi (1954) as "volcanic-pyroclastic Pampean-Patagonian association," dominate most of the Argentinean margin. They are transferred to the sea from the mountain and peri-mountain regions to the plains and coast by eolian, fluvial, and glacial activity along with intricate multicyclic processes. Wave erosion of old continental sequences outcropping at the coast is the main factor as a sediment supplier to the shelf as it was also during lower-than-today sea-level positions (Gaiero et al., 2003; Violante and Parker, 2004; Isla and Cortizo, 2014; Violante et al., 2014). Another

sediment source for the local concentration of gravels in the slope and rise are the icebergs that supplied large amounts of ice-rafted debris (containing rock fragments from Antarctica) during glacial times and concentrated in some places by bottom flows (Krinsley and Biscaye, 1973; Muñoz et al., 2012; Bozzano et al., 2021). Part of this ice-rafted debris reached the present Uruguayan margin (Franco-Fraguas et al., 2014).

The Plata basin-sourced sediments are mainly supplied by the Río de la Plata along the Uruguayan coast, reaching the southern

Brazilian shelf (Mathias et al., 2014). A mixture of different sources is observed in these sediments (Manassero et al., 2008), reflecting distinct sources. These sources include the metamorphic rocks from the Brazilian shield, basalts from the upper



Paraná basin, and sedimentites/volcanic rocks supplied by the Pilcomayo and Bermejo rivers basins from the northern Argentine/Bolivian Andes and the Chaco plains.

Sediments from the Brazilian craton are originated from sources located to the north of the area of study (Anjos et al., 2007; Cruz et al., 2018) and transported by the Brazil Current (Viana et al., 1998; de Mahiques et al., 2004). The presence of a 1,000-km long coastal mountain range (Serra do Mar), extending from 28ºS to 23ºS, hampers the direct input of sediments to the adjacent margin (Cogné et al., 2011; de Mahiques et al., 2017).

**2.3 Ocean Circulation**

The Southwestern Atlantic margin is characterized by complex hydrography (Matano et al., 2010). It presents two main oceanographic boundaries, the Subtropical Shelf Front (STSF), as the shelf extension of the Brazil – Malvinas Convergence (BMC) (Piola et al., 2000; Severov et al., 2012), and the less-studied Santos Bifurcation (SB) (Boebel et al., 1997; Boebel et al., 1999a). It is also influenced by the Rio de la Plata (RdlP), which discharges freshwater from the second-largest hydrographic basin in South America, with an average value of 22,000 $m^3$ $s^{-1}$ (Framiñan and Brown, 1996). This regional circulation system experiences seasonal latitudinal shifts in response to wind regimes (Schmid et al., 2000; Piola and Matano, 2001; Piola et al., 2018).

At the BMC, centered at 37-39°S (Maamaatuaiahutapu et al., 1992), the southward-flowing Brazil Current (BC) encounters the northward-flowing Malvinas Current (MC) (Schmid and Garzoli, 2009), displacing water masses with contrasting thermohaline characteristics. The BC is a baroclinic boundary current that concentrates its main flow up to 500 m water-depth, carrying the Tropical Water (TW) at the surface (Emilsson, 1961; Palma et al., 2008), and the South Atlantic Central Water (SACW) at pycnocline levels (Emilsson, 1961; Signorini, 1978). The MC is a strong barotropic boundary current that advects the Subantarctic Water (SAW) at the surface (Spadone and Provost, 2009), and the Antarctic Intermediate Water (AAIW) at intermediate levels (Tomczak and Godfrey, 1994a, b).

At the BMC, water masses are displaced eastwards as the basin-wide Anticyclonic Atlantic Subtropical Gyre (Boebel et al., 1997; Boebel et al., 1999a; Schmid et al., 2000; Núñez-Riboni et al., 2005; Legeais et al., 2013). At the intermediate levels of the westward flow of the gyre, the water reaches the South American margin near 28ºS, where it splits into two branches, inducing the formation of the Santos Bifurcation (Boebel et al., 1999a; Legeais et al., 2013). From the bifurcation, one-quarter of the transport at 40ºW flows northward along the margin (mainly between the 800 and 1,200 m isobaths), forming the Intermediate Western Boundary Current (IWBC) (Fernandes et al., 2009; Biló et al., 2014). About three-quarters flow to the south, following the BC, until its recirculation in the BMC (Schmid et al., 2000; Piola and Matano, 2019). This configuration leads to an overall southward flow on the outer shelf and outer to middle slope, from 28ºS up to the BMC.

Concerning deep circulation, the southward flow of the North Atlantic Deep Water (NADW) (Sverdrup et al., 1942), transported from northern hemisphere high latitudes by the Deep Western Boundary Current, stratifies at 2,000 – 3,000 m water depth. The NADW flows between two northward-flowing branches of the Circumpolar Water (i.e., Upper and Lower


Circumpolar Deep Water). The abyssal circulation (> 3,500 m) is dominated by the Antarctic Bottom Water (AABW), which

is partially trapped in the Argentine Basin (Tarakanov and Morozov, 2015).

This large-scale oceanographic process is closely related to shelf circulation (Matano et al., 2010). Over the shelf, the extension of the BMC, known as the Subtropical Shelf Front (STSF), separates Subtropical Shelf Waters (STSW, formed by the mixture of the TW and SACW) and Subantarctic Shelf Waters (SASW) (Piola et al., 2000). This narrow and sharp front extends between 32°S at 50 m of water column depth and 36°S over the shelf break, and its distribution appears stable throughout the

year (Piola et al., 2000; Berden et al., 2020). At the STSF, the main branch is mixed with waters transported by the BC and exported offshoreward along with the BMC. A secondary branch is diluted with the PPW and TW and returns along the shelf (Berden et al., 2020).

At the surface, the low-salinity RdlP plume flows northward along the inner Uruguayan continental shelf during the austral winter. In the summer and during El Nino events, the plume remains off the RdlP mouth and extends along the entire upper

continental margin (Piola et al., 2000; Piola et al., 2005; Möller et al., 2008).

## 3 Materials and Methods

In this study, the samples were organized in five distinct sectors, with names corresponding to the Santos, Pelotas, and Punta del Este marginal basins, RdlP estuary, and the Argentinean margin. Due to the small number of samples, the sediments from Argentina were not divided into the corresponding sedimentary basins (Figure 1). Geographic coordinates and water depth of

the samples are presented in the Supplementary Material file.

Besides 83 new sediment samples, we gathered information on the previous works by Basile et al. (1997), de Mahiques et al. (2008), and Franco-Fraguas et al. (2016). We also included the εNd values of the shallowest (Late Holocene) core samples presented by Lantzsch et al. (2014). The methodology used in those ancillary papers is described in the original references. The new samples were collected in distinct surveys onboard the research vessels Alpha Crucis (Brazilian margin), Miguel

Oliver, Capitán Saldaña, and Sarmiento de Gamboa (Uruguayan margin), using box-corers and multiple-corers. Only the superficial samples (1st centimeter) of each core were used in this work.

The Nd and Pb isotopic analyses of the lithogenic fraction were carried out at the Geochronological Research Centre of the University of São Paulo, Brazil.

All chemical procedures were performed in class 10,000 cleanroom equipped with laminar flow hoods class 100. All reagents

were purified before use. Water was distilled and then purified on a Milli-Q System (®Millipore Corporation) ('ultrapure' water - "Type 1"). The acids were purified in sub boiling distillers (DST-1000, ®Savillex) and sub boiling stills (®Savillex) at low temperatures.

Sediment powder (70 mg) was dissolved with HF, HNO3, and HCl acids. Dissolution was done on a MARS-5 microwave oven. Both Pb and Nd were purified by the ion-exchange technique. The first stage of ion-exchange chromatography involves

separating Pb from the other matrix elements using columns packed with anion exchange AG1-X8, 200-400 mesh (Biorad)





resin. After Pb collection, the remaining solution is dried out, and the residue retaken for separation of the rare earth elements using RE resin (EIChroM Industries Inc.) from the bulk solution. Nd was then separated using Ln resin (EIChroM Industries Inc.).

Pb isotopic compositions were measured on a Finnigan MAT 262 Mass Spectrometer. Samples were loaded on Re filaments

with $H_3PO_4$ and silica gel. Every single analysis consisted of 60 ratio measurements. The Pb ratios were corrected for mass fractionation of 0.13%/amu based on repeated analysis of the NBS-981 standard (Pb/Pb = 16.893 ± 0.003; Pb/Pb = 15.432 ± 0.004, and Pb/Pb = 36.512 ± 0.014; n = 11), which yielded mass discrimination and fractionation corrections of 1.0024 (Pb/Pb), 1.0038 (Pb/Pb) and 1.0051 (Pb/Pb). The combination of these uncertainties and within-run uncertainties are typically 0.15%–0.48% for Pb/Pb, 0.13%–1.07% for Pb/Pb and 0.10%–0.45% for Pb/Pb, all at the $2\sigma$ (95%) confidence level. The total Pb

blank contribution, <1 ng, is negligible.

The Nd analyses, here reported as εNd, were prepared by standard methods by the analytical procedures described by Sato et al. (1995) and Magdaleno et al. (2017), involving the removal of calcium carbonate, HF–HNO$_3$ dissolution plus HCl cation exchange using a Teflon Powder column to separate REE. No visible solid residues were observed after dissolution. Samples with incomplete dissolution were discarded.

Nd determinations were performed on a Thermo Neptune Plus ICP-MS. Nd isotopic ratios (Nd/Nd) were normalized to the value of $^{146}Nd/Nd = 0.7219$ (DePaolo, 1981) and Nd/Nd = 0.512103 of the JNDi-1 standard (laboratory average of the last 12 months). Usually, a single analysis consisted of 60 measurements of Nd. The Nd/Nd mean average of the JNDi standard during the analyses was 0.512095 ± 0.000007 (n = 3) and 0.512096 ± 0.000005 between July and November of 2013 (n = 56). The daily average of Nd/Nd of the JNDi-1 standard was 0.512101±0.000002 (n=18). The analytical blank during the analyses

varied from 51 and 53 pg.

The parameter εNd was calculated as follows:

εNd = ((Nd/Nd$_{sample}$/Nd/Nd$_{CHUR}$)-1) * $10^4$, where Nd/Nd$_{CHUR}$ = 0.512638 (Jacobsen and Wasserburg, 1980).

Statistical analyses were performed using the software PAST (Palaeontological Statistics), version 4.05 (Hammer et al., 2001). To recognize the distinct isotopic domains over the study area, we applied a procedure similar to the one proposed by Walling

(2013), Miller et al. (2015b), and Palazon and Navas (2017). First, a Kruskal-Wallis non-parametric analysis of variance was applied for each variable, followed by a Mann-Whitney pairwise post-hoc test to identify which variables presented statistically significant differences. Finally, a Discriminant Analysis with standardized values was used to determine the correct classification for the previously assigned groups.

To support the interpretation of the geochemical data distribution, we analyzed the output of the LLC2160 simulation, a global

1/24º forward run of the Massachusetts Institute of Technology General Circulation Model (MITGCM) that was spun up from Estimating from Circulation and Climate of the Ocean (ECCO). ECCO is an ocean reanalysis, which assimilates millions of observations, starting in 1992. With 90 vertical levels and a horizontal resolution of about 4 km in the South American margin, the LLC2160 simulation resolves the main ocean circulation features on the continental slope and shelf of the southwestern



Atlantic. Our analysis focuses on a 12-month average spanning September 2011 through August 2012. Details of the

simulation, including a description of the spin-up hierarchy and forcing, are available in Chen et al. (2018).

We used annual-mean fields of LLC2160 simulation to identify two key features: the Santos Bifurcation (SB) and the

Subtropical Shelf Front (STSF). The SB is recognized as the region on the continental slope where the flow within the AAIW

depth range (550-1400 m) is negligible. Specifically, we search on different isobaths ranging from 500 m to 1500 m for the

region where the AAIW flow is weaker than 0.01 m/s.  We emphasize that the SB is not a stagnation point where the flow is

zero but a shadow zone that spans nearly 100 km, wherein the intermediate flow is feeble (see the schematic SB in Figure 1).

In our discussion below, we present the mean position and the latitudinal extension of the SB as a function depth.

To identify the mean position of the STSF, we searched for the local maximum of the potential temperature gradient, which is

a very distinct feature on the northern Argentina/southern Brazil shelf. We compute the potential temperature gradients at 40

m to avoid contamination by RdlP water (e.g., Piola et al., 2008). When applied to the LLC2160 output using seasonal averages,

our method yielded frontal locations consistent with those identified by applying the isothermal criteria at 40 m proposed by

Piola et al. (2008). In the yearly fields, the front follows approximately the 14 ºC isotherm.

## 4 Results

The results of isotopic analyses are presented in the Supplementary Material and summarized in the box-plots shown in Figure

2.  We also present the latitudinal variation of each isotope (Figure 3).

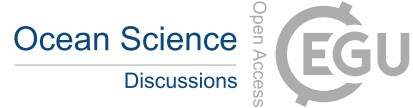

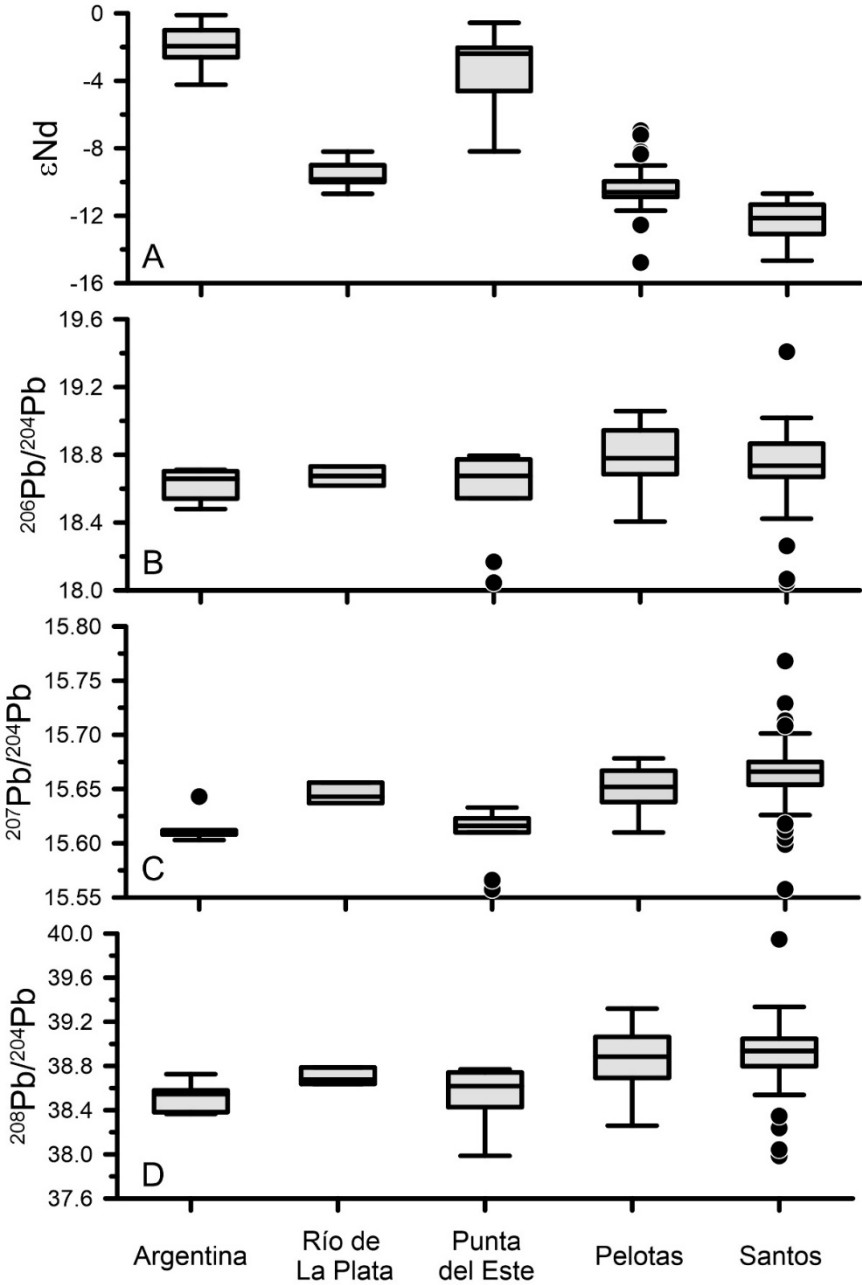

**Figure 2. Box-plots of the distributions of (A) εNd, (B) $^{206}Pb/^{204}Pb$, (C) $^{207}Pb/^{204}Pb$, and $^{208}Pb/^{204}Pb$. Outliers are shown as dots**





**Figure 03. Latitudinal variations of (A) εNd, (B) $^{206}$Pb/$^{204}$Pb, (C) $^{207}$Pb/$^{204}$Pb, and $^{208}$Pb/$^{204}$Pb. Symbols: Argentina (blue dots), Punta del Este Basin (red squares), Río de la Plata (brown X), Pelotas Basin (black triangles), and Santos Basin (green stars).**





εNd values show a northward trend to less radiogenic values, varying from -0.1 (Argentina) to -17.1 (Santos Basin) (Figure

3a). The latitudinal variation of the Pb isotopes is less clear but still visible for Pb/Pb and Pb/Pb (Figures 3c and 3d). On

average, the Argentina sector presents the highest εNd average values (-2.1±1.3), and lowest Pb/Pb, Pb/Pb, and Pb/Pb average

values (18.620±0.104, 15.615±0.016, and 38.520±0.149, respectively). On the other hand, the Santos sector shows the lowest

εNd (-12.0±1.1) and highest Pb/Pb and Pb/Pb average values (15.664±0.008 and 38.909±0.016, respectively). Values of Pb/Pb

did not show any evident latitudinal trend.

For Pb/Pb, the values range from 18.045, on the Punta del Este sector, to 19.409, on the Santos Basin shelf. Pb/Pb values

range from 15.558 to 15.768 in the same areas. Finally, Pb/Pb values vary from 37.986 to 39.949, also in the same sectors.

One sample, located on the shelf of the Punta del Este sector (35.0°S – 53.2°W, 47 m), provided the least radiogenic values for

the three Pb isotopes. On the other hand, a sample located at 27.1°S – 48.2°W, at 56 meters of water depth, provided the highest

Pb/Pb and Pb/Pb. The highest value of Pb/Pb was obtained in a sample located at 25.2°S – 46.1°W, at a water depth of 97

meters.

From the Kruskal-Wallis analysis, we observe that except for Pb/Pb, the variables show significant differences among the

compartments, thus proceeding with the Discriminant Analysis. Furthermore, the Mann-Whitney analysis allowed us to

recognize the pairwise differences among the other variables (Table 1). Finally, it is to be noted that sediments from Argentina

showed statistically significant differences with all of the variables analyzed, suggesting that they are distinct from those

located towards the North. On the other hand, sediments from the Rio de la Plata are statistically similar to those from the

Pelotas sector for all of the variables.





**Table 1.  p values of the Mann-Whitney pairwise test. Statistically, significant differences are highlighted in blue**

| $\varepsilon$Nd | Argentina | Rio de la Plata | Punta del Este | Pelotas |
|---|---|---|---|---|
| Argentina | | | | |
| Rio de la Plata | 0.00 | | | |
| Uruguay | 0.09 | 0.00 | | |
| Pelotas | 0.00 | 0.32 | 0.00 | |
| Santos | 0.00 | 0.00 | 0.00 | 0.00 |
| | | | | |
| $^{206}$Pb/$^{204}$Pb | Argentina | Rio de la Plata | Punta del Este | Pelotas |
| Argentina | | | | |
| Rio de la Plata | 0.21 | | | |
| Uruguay | 0.59 | 0.59 | | |
| Pelotas | 0.02 | 0.31 | 0.08 | |
| Santos | 0.08 | 0.69 | 0.37 | 0.08 |
| | | | | |
| $^{207}$Pb/$^{206}$Pb | Argentina | Rio de la Plata | Punta del Este | Pelotas |
| Argentina | | | | |
| Rio de la Plata | 0.07 | | | |
| Uruguay | 0.42 | 0.29 | | |
| Pelotas | 0.00 | 0.06 | 0.00 | |
| Santos | 0.00 | 0.02 | 0.00 | 0.01 |
| | | | | |
| $^{208}$Pb/$^{204}$Pb | Argentina | Rio de la Plata | Punta del Este | Pelotas |
| Argentina | | | | |
| Rio de la Plata | 0.04 | | | |
| Uruguay | 0.59 | 0.29 | | |
| Pelotas | 0.01 | 0.11 | 0.02 | |
| Santos | 0.00 | 0.02 | 0.01 | 0.47 |


The two first axes of the Discriminant Analysis account for 99.72% (99.09% for axis 1) of the total variance considering the standardized values of $\varepsilon$Nd,Pb/Pb, and Pb/Pb (Figure 4).  It is possible to recognize that samples from Argentina are detached



from the other sectors. On the other hand, samples from Pelotas Basin show a transitional character between Santos Basin, on
one side, and Río de La Plata and Punta del Este Basin.



**Figure 4. Scatter plot of the samples according to the two first axes generated from the Discriminant Analysis**

Graphical outputs of the LLC2160 model are presented for both the Santos Bifurcation (Figure 5) and Subtropical Shelf Front
(Figure 6). The Santos Bifurcation is identified as the region of maximum horizontal velocity divergence at the AAIW level,
identified in Figure 5A close to 26ºS. The visualization based on horizontal fields is more complicated but still visible as the
sector with velocities close to 0 m/s (Figure 5B). The Subtropical Shelf Front (Figure 6A) is identified as a maximum
subsuperficial temperature gradient (Figure 6B). Vertically it is well marked below the 30 m isobath (Figure 6C).



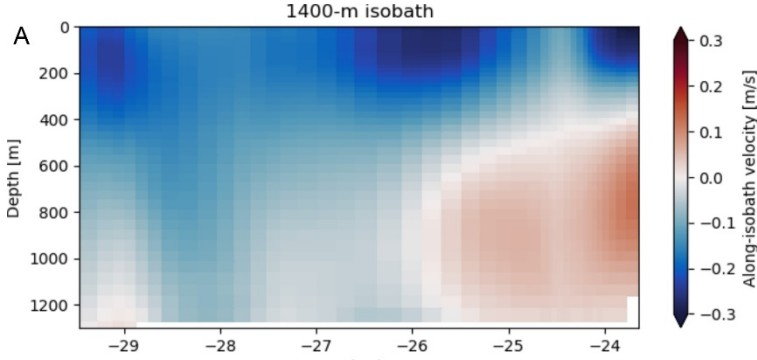

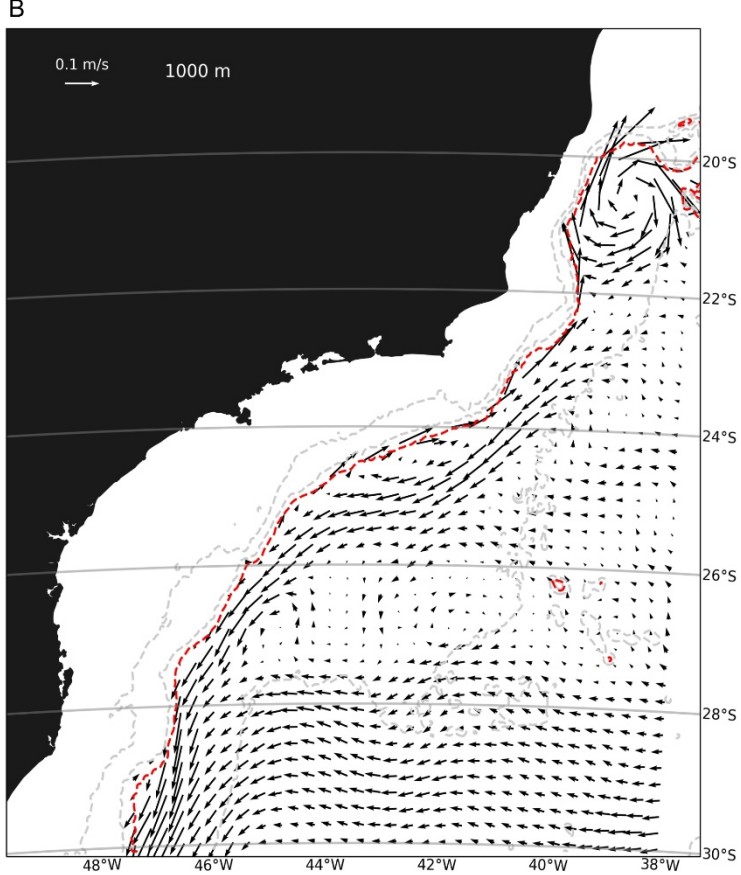

**Figure 5. Graphical outputs of the LLC2160 focused on the Santos Bifurcation. 6A: the bifurcation is identified as the white zone located around 26ºS between the 800 and the 1200 isobath. 6B: the bifurcation is recognized as the zone of velocities tending to 0 cm/s at 26ºS – 44ºW.**

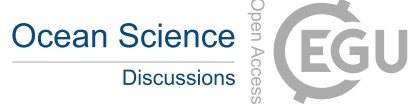

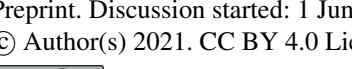

**Figure 6. Graphical outputs of the LLC2160 focused on the Subtropical Shelf Front. 6A: Variations of temperature in the zone of the front. 6B: Temperature gradient; the front is recognized as the zone of maximum gradient (darker colors). 6C: Vertical transect at 35°S showing the location of the Subtropical Shelf Front (STSF) as the gradient of the Subantarctic Shelf Water (deep blue) and the Subtropical Shelf Water (light green)**





## 5 Discussion

Recognizing the role of circulation on the deposition of sediments requires associating the sedimentary provinces with potential
source areas of sediments. Indeed, radiogenic isotopes are considered good sediment source fingerprints (Owens et al., 2016).
Two seminal papers, by Goldstein et al. (1984) and Bayon et al. (2015), used Nd isotopes and other proxies from the world´s
rivers and provided the basis for the comprehension of distribution detrital Nd in the world´s oceans. In the South Atlantic
case, an important outcome provided by Beny et al. (2020) is the summary of Nd, Pb, and Sr signatures provided for the
potential sources and circulation in the area.

The scatter plots of εNd versus Pb/Pb and εNd versus Pb/Pb (Figure 7) allow identifying distinct patterns that can be associated
with different sources for the sediments of the Southwestern Atlantic margin. Together with the samples, we considered the
average and standard deviation values of the following potential sources:

a.      Sediments collected from the Antarctic Peninsula and Western Antarctica, between 2600 and 3800 m water depth
(Roy et al., 2007);

b.      Topsoils, river bed sediments, and eolian dust from Patagonia (Gaiero et al., 2007);

c.      Topsoils and river bed sediments from Southern Patagonia (Khondoker et al., 2018);

d.      Clay and silt fractions from the Chubut River (Patagonia) (Bayon et al., 2015);

e.      Bulk sediment from Paraná River (Goldstein et al., 1984);

f.      Suspended sediments from Paraná and Uruguay rivers, the two main watercourses that form the RdlP (Henry et al.,
325   1996);

g.      Nd and Pb values from low-Ti basalts from the southern Paraná Igneous Province (Barreto et al., 2016; Melankholina
and Sushchevskaya, 2018);

h.      River sediments draining Proterozoic low- to middle-rank metamorphic rocks from southeastern Brazil (river mouth
located at 24.68ºS – 047.42ºW) (Moraes et al., 2004);

i.      Neoarchean and Proterozoic metasediments from the coastal region of southeastern Brazil (Ragatky et al., 2000);

j.      Proterozoic granites from the coastal region of southeastern Brazil (Mendes et al., 2011).


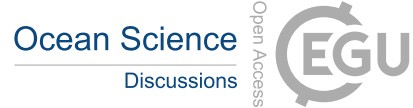

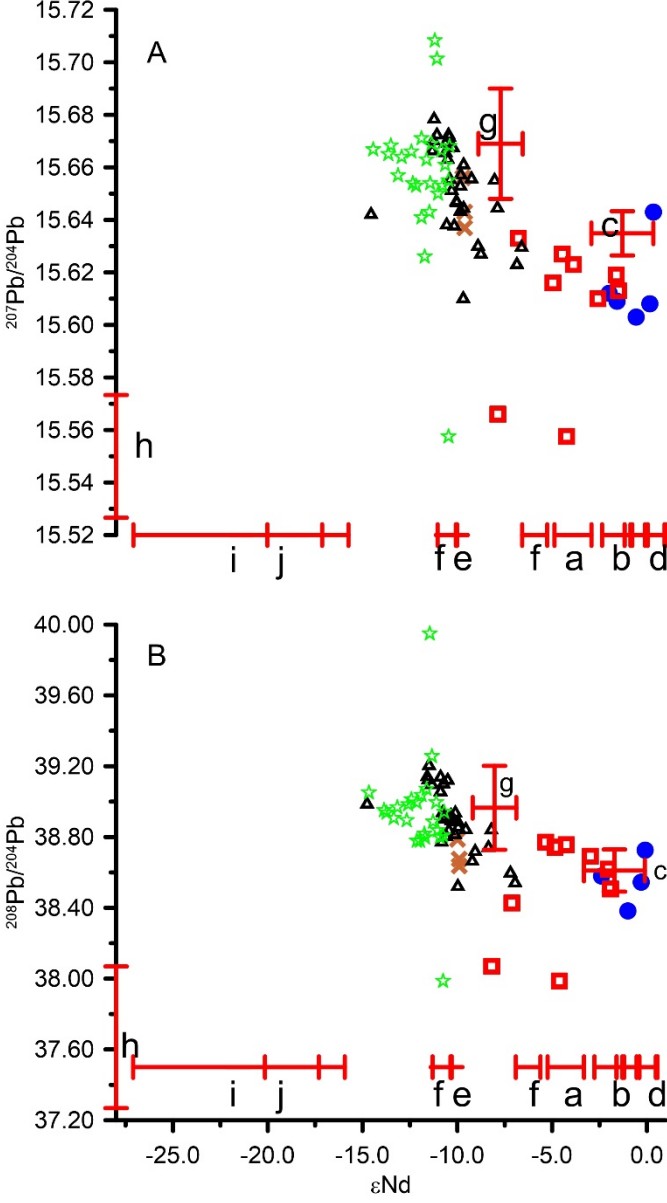

**Figure 7.** Scatter plots of εNd versus $^{207}$Pb/$^{204}$Pb and εNd versus $^{208}$Pb/$^{204}$Pb of the samples and potential sources. Letters: a. Sediments collected from the Antarctic Peninsula and Western Antarctica, between 2600 and 3800 m water depth (Roy et al., 2007); b. Topsoils, river bed sediments, and eolian dust from Patagonia (Gaiero et al., 2007); c. Topsoils and river bed sediments from Southern Patagonia (Khondoker et al., 2018); d. Clay and silt fractions from the Chubut River (Patagonia) (Bayon et al., 2015); e. Bulk sediment from Paraná River (Goldstein et al., 1984); f. Suspended sediments from Paraná and Uruguay rivers, the two main watercourses that form the RdlP (Henry et al., 1996); g. Nd and Pb values from low-Ti basalts from the southern Paraná Igneous Province (Barreto et al., 2016; Melankholina and Sushchevskaya, 2018); h. River sediments draining Proterozoic low- to middle-rank metamorphic rocks from southeastern Brazil (river mouth located at 24.68oS – 047.42oW) (Moraes et al., 2004); i. Neoarchean and Proterozoic metasediments from the coastal region of southeastern Brazil (Ragatky et al., 2000); j. Proterozoic granites from the coastal region of southeastern Brazil (Mendes et al., 2011);





Sediments from Argentina and part of the Punta del Este Basin present isotopic signatures similar to the values obtained for Patagonia (Gaiero et al., 2007; Bayon et al., 2015; Khondoker et al., 2018). The deepest samples of the dataset, located in the Punta del Este basin, at water depths between 2378 and 4066 meters, present εNd values between -4.26 and -5.33. These values are compatible with those of the Antarctic Peninsula and West Antarctica (Roy et al., 2007). They can indicate a provenance

of sediments via the flow of the Upper- and Lower- Circumpolar Deep-water masses (UCDW and LCDW, respectively) (Beny et al., 2020) or even from ice-rafted debris. The rest of the samples from the Punta del Este basin, situated on the shelf, might represent a mixture of Patagonian and Río de la Plata sediments.

Most of the samples from the Pelotas Basin are under the influence of the RdlP. Apart from Nd and Pb isotope values, other independent proxies confirm the Plata Plume Water (PPW) as a source of the sediments to the southern Brazilian margin

(Pelotas Basin) and part of the southeastern margin (Santos Basin). Campos et al. (2008) and Nagai et al. (2014a) used clay mineralogy to indicate the transport of sediments from the Río de la Plata sediments to the North. Also, the maps presented by Govin et al. (2012) show similarities in the ln(Ti/Al) and ln(Fe/K) between the Uruguayan and southern Brazilian upper margin. Finally, Mathias et al. (2014) used magnetic properties of sediments in a core located at the latitude of 25º30'S to recognize the influence of the Río de la Plata on the southern Brazilian shelf since 2 cal kyr BP. Finally, in a study that included

the analysis of potential source rocks from the continent, Mantovanelli et al. (2018) confirmed the contribution of the Paraná Basin basalts along the Holocene off the southern Brazilian shelf (27ºS). The authors observed a remarkable change to less radiogenic Nd in the sedimentary column further North (23ºS).

The samples from Santos Basin present lower radiogenic Nd and higher radiogenic Pb values, thus indicating a Pre-Cambrian source, as Mantovanelli et al. (2018) stated. Nevertheless, the values obtained for Pb isotopes differ significantly from those

reported by the literature for the Precambrian metasediments and granites of the southeastern Brazilian coast (Ragatky et al., 2000; Moraes et al., 2004; Mendes et al., 2011). A possible explanation for this discrepancy lies in the fact that the input of sediments from the adjacent coast is hampered by the presence of the Serra do Mar mountain chain, which limits the development of expressive drainage basins in the area. In this sense, we cannot discard the possibility that a significant part of sediments that presently cover the shelf and upper slope of the Santos basin is originated further north and transported by

the Brazil Current and derived shelf dynamics (Castro and Miranda, 1998; Silveira et al., 2017).

Another point of view can be obtained from the classed plot according to latitude and water depth (Figures 8 and 9). Figures 8A and 8B present the latitudinal and bathymetric variability of εNd and Pb/Pb, respectively, between 35ºS and 55ºS. Three main groups are recognized in this sector. The lowest εNd values are associated with the presence of the PPW and extend up to about 80 m water depth (Figure 8A). The more Nd radiogenic sediments (-2.0 < εNd < 0.0) prevail on the Argentinean

shelf and slope and are limited by the 1,000 m isobath; when considering the values between -1.0 and 0.0, the bathymetrical limit is 200 m. Finally, values of εNd between -4.0 and -6.0 occur below the 2,500 m isobath, areas influenced by the LCDW.




A less evident but still present bathymetric control for $^{207}$Pb/$^{204}$Pb (Figure 8B), the higher values occur in water depths shallower than 80 m.

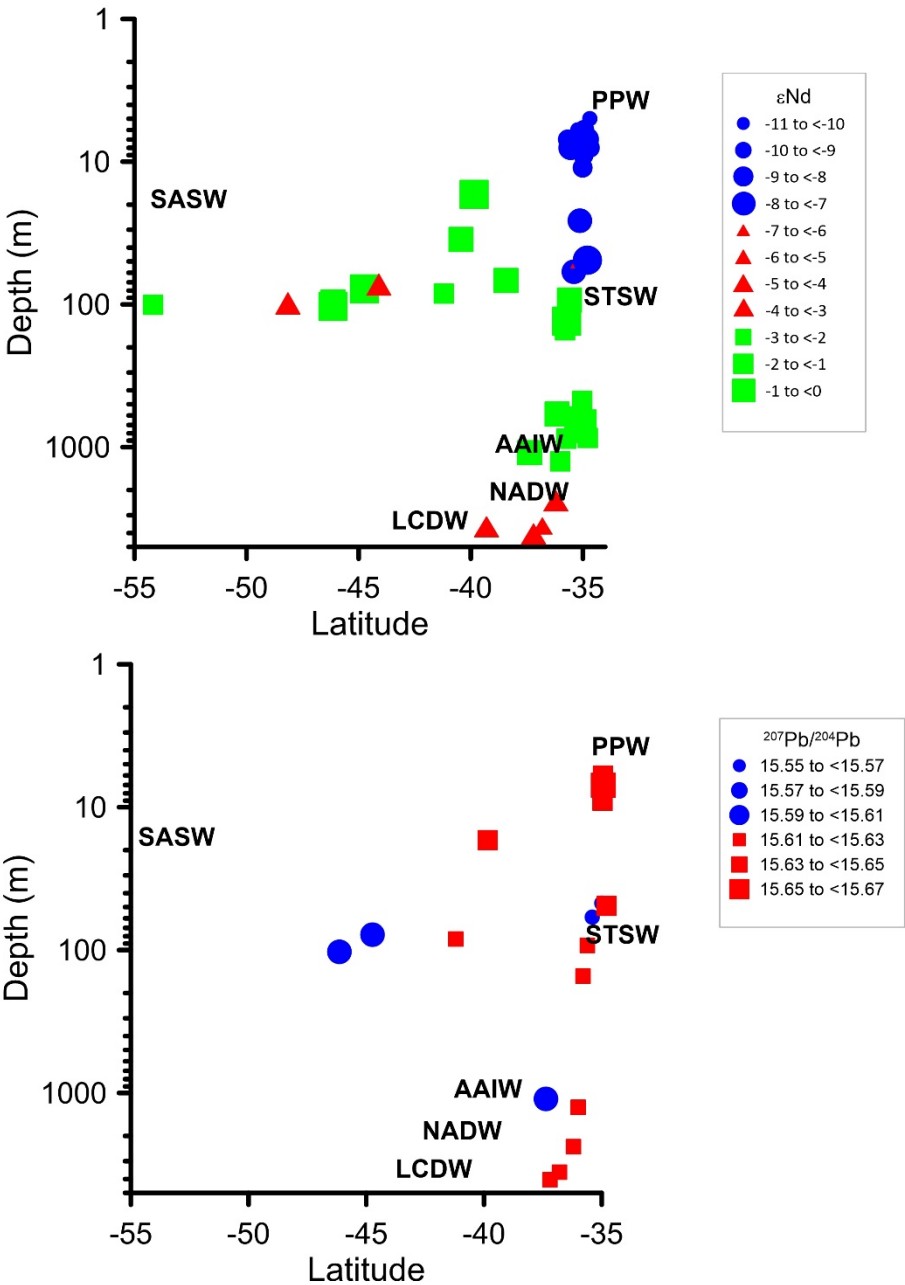

**Figure 8. Latitudinal and bathymetric variability of εNd (above) and Pb/Pb (below) in the sector between 35ºS and 55ºS. Water masses: Plata Plume Water (PPW), Subantarctic Shelf Water (SASW), Subtropical Shelf Water (STSW), Antarctic Intermediate Water (AAIW), North Atlantic Deep Water (NADW), and Lower Circumpolar Deep Water (LCDW). Vertical axis in Log$_{(10)}$ scale.**



Figure 9 presents the bathymetrical variations of εNd between 36ºS and 23ºS and positions of the STSF (dashed line) and SB (dashed line with horizontal bars) identified in the LLC2160 simulation. The positions of both oceanographic features in the LLC2160 simulation are broadly consistent with previous studies (e.g., Boebel et al., 1999a; Boebel et al., 1999b; Piola et al., 2008). The LLC2160 output allows us to present, together with εNd, the bathymetrical variations of those features. As observed, there are clear distinctions in the signature corresponding to both fronts. In the case of the STSF, our findings

confirm the conclusions previously stated by Franco-Fraguas et al. (2014). The STSF presents only minor seasonal variations, and its control is probably related to the interaction between the RdlP plume and the subsurface water masses distribution. During austral summer, the strong stratification (Möller et al., 2008) inhibits RdlP sedimentation southward of the STSF. During austral winter, the northeastward RdlP plume promotes the offshore displacement of subtropical waters (Möller et al., 2008), enabling the deposition of fine sediments on the shelf north of the STSF. On the upper and middle slope, the southward

displacement of the Brazil Current, as well as of the recirculated AAIW, likely limits RdlP sedimentation (Schmid et al., 2000). There is a clear difference in the εNd signatures in the intermediate zone, at about 34ºS – 35ºS; this boundary might represent the northernmost limit of the BMC (Benthien and Müller, 2000; Pezzi et al., 2009).





**Figure 9. Bathymetric variation of the values of εNd, between 24ºS and 36ºS and its relationship with the Santos Bifurcation (SB) and the Subtropical Shelf Front (STSF). The SB is presented as a variable zone, where the current velocity is lower than 0.1 m/s. Abbreviations: Antarctic Intermediate Water (AAIW), South Atlantic Central Water (SACW), Tropical Water (TW), Subantarctic Shelf Water (SASW), Plata Plume Water (PPW).**

This integrated analysis allows us to interpret that there is no transport of sediments from the Argentinean sector to the southern Brazilian margin. On the other hand, based on the same analysis, we can confirm that sediments from the Rio de la Plata reach, at least partially, the Santos sector, i.e., to the north of 28ºS.

Concerning the SB, there is a clear distinction in isotopic signatures below the 500 m isobath, less radiogenic Nd prevailing to the north of the bifurcation. We thus argue that both STSF and SB are also limiting distinct geochemical provinces on the southwestern Atlantic margin.





A conceptual model of sediment source and transport is shown in Figure 10. The Argentinean and part of the Uruguayan upper margins are covered by Andean-Patagonian sediments, which are redistributed by the shelf circulation and Malvinas Currents. The STSF and BMC block the transport of these sediments to the north. This finding corroborates with Hernández-Molina et al. (2016) and Franco-Fraguas et al. (2016). Sediments located deeper than the 2,000 m isobath present an Antarctic signature, transported either by the bottom circulation (UCDW and LCDW) or as ice-rafted debris. Sediments from the Río de la Plata estuary advance along the inner shelf towards southern Brazil, and a mixture of Pelotas and Santos signatures are observed between 28ºS and 30ºS. This mixture is visible in the scatter plot presented in Figure 4, in which sediments of Pelotas Basin constitute a mixture of distinct populations, i.e., Santos Basin and Río de la Plata. Finally, sediments located northward of 27ºS are originated from the Precambrian rocks that dominate the coastal domains off SE and E Brazil, being mainly transported by the intense flow of the BC. Limited input comes from the small rivers that drain the mountainous areas of the Serra do Mar.



Figure 10. Conceptual model of sediment sources and transport along the Southwestern Atlantic margin. The size of the arrows corresponds to the qualitative importance of the transport
**Conclusions**

In this paper, we use Nd and Pb radiogenic isotopes to recognize the role of ocean circulation in the sediment distribution of
the southwestern Atlantic margin.

Andean and continental Patagonian sediments are the primary source for the deposits of the Argentinean and Uruguayan
shelves, while the lower slope is more influenced by more distant sources, such as the Antarctic Peninsula. Nevertheless,
sediments on the shelf and upper slope are carried by the flows of the SASW and AAIW, while the UCDW and LCDW
transport sediments from the lower slope.

The Río de la Plata is recognized as the primary influencer of the sediments off southern Brazil up to the 27ºS parallel. The
sediments are transported northwards by the PPW, which is transported by a wind-driven current. A mixture of sediments
from the PPW and the north is transported towards the slope between 34ºS and 28ºS.

Finally, Pre-Cambrian terrains are the primary sources of the sediments deposited further north. They are originated from
rivers located northward of the area of study and, on a smaller scale, by the small drainages that face the ocean in the Serra do
Mar region.

We propose that the main oceanographic boundaries of the southwestern South Atlantic margin, i.e., the Subtropical Shelf
Front and the Santos Bifurcation, act as limits of distinct geochemical provinces. Thus, the boundary represented by the STSF
extends towards deeper areas along with the Brazil – Malvinas Confluence.

**Data availability**

All of the data used in this paper is presented as Supplementary Material

**Acknowledgments**

The authors acknowledge the crew of R.V. Alpha Crucis and participants of research cruises Mudbelts I and II and Talude I
and II for helping during the sampling surveys. This work is a contribution to the Grupo de Investigación en Ciencia y
Tecnología Marina (CINCYTEMA) as well as to the MOU between the Oceanographic Institute of University of São Paulo
(Brazil) and the Facultad de Ciencias of the Universidad de La Republica (Uruguay), institutions to whom the authors are
indebted.

**Funding**

This work was financially supported by the São Paulo Science Foundation (FAPESP), for grants 2010/06147-5, 2014/08266-
2, 2015/17763-2, 2016/22194-0, and 2019/00256-1. MM de M acknowledges the Brazilian Council of Scientific Research
(CNPq) for the Research Grant 300962/2018-5.



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
