# Peer review of "Control of oceanic circulation on sediment distribution in the Southwestern Atlantic margin (23°S to 55°S)"

_Ocean Science, 2021_

## Referee Comment (RC1)

**Review of de Mahiques et al "New insights of the influence of ocean circulation on the sedimentary distribution in the Southwestern Atlantic margin (23 S to 55 S) based on Nd and Pb isotope fingerprinting" in Ocean Science**

This paper published a large set of Nd and Pb isotope data for sediments from five distinct locations, the Santos, Pelotas, Punta del Este, R ó de la Plate and Argentinean margin. The authors try to use Nd and Pb isotope data to discuss the sources of the sediments and thus the transport model from provenance to the deposition area. Although the isotope dataset is worth publishing, there are a number of problems with the language expression and data interpretation, which will require major revision or complete rewriting of the manuscript.

I found the paper is difficult to follow because of the language. The authors need to polish the language in this paper.

In addition, I found most of the ideas in this paper has been discussed in the published paper of **de Mahiques et al., 2008, Marine Geology.** The authors need to highlight new insights about the sediment sources and ocean circulations.

I still have detailed reviews as following:

21 Not true. There is latitudinal trend for Nd isotopes, but not for Pb isotopes.

40 "Rare Earths" should be "Rare Earth Elements" or "REEs"

260-264 Pb/Pb? Please correct the ratios in this sentence and the flowing sentences through all manuscript

347 The sediment samples from Punta del Este basin also have Nd and Pb isotope ratios between Antarctic sediments (a endmember) and Paran á Igneous Province (g endmember). How to exclude this provenances?

353-363 The authors try to attibute the Nd and Pb isotope data of Pelotas samples to the influence from R  $\acute{b}$  de la Plate Plume water. But the Pelotas samples display larger range of Nd and Pb isotope compositions than R  $\acute{b}$  de la Plate samples. It is evidence that the Pelotas samples were effected by other sediment provenances. However, the authors did not discuss this point. In

addition, the Pelotas sediment samples are obviously located in the area of Brazil Current (Fig. 1), but the authors did not discuss the possible influence from Brazil Current.

363-370 It could not convince me that the Santos samples are related to Precambian metamorphic rocks and granites of the Southeastern Brazilian coast. The Nd and Pb isotope data of the Santos sediments and Precambian metamorphic rocks as well as granites of the Southeastern Brazilian coast (h, i and j end members) are obviously distinct. The following transport model of the Precambrian Brazilian cannot convince me as well.

418-419 It is not true. I observed that the Pelotas samples display mixture characteristics between Santos samples and Punta del Este samples, not R ó de la Plate samples.

420-421 It is not true. The Santos samples did not have similar Nd and Pb isotope compositions compared to the Precambian rocks (Fig. 7).

What is the Nd and Pb isotope compositions of Two Argentina samples from Brazil-Malvinas confluence (BMC)? Did they inherit the isotope compositions from Malvinas Current or Brazil Current?

Fig. 1 what is thin red line which is basically parallel with thick red line in this figure?

---

## Author Comment (AC1)

To the Editor of Ocean Science
Ref. 1st review of the manuscript
"New insights of the influence of ocean circulation on the sedimentary distribution in the Southwestern Atlantic margin (23ºS to 55ºS) based on Nd and Pb isotope fingerprinting"

Dear Sirs,

First of all, we would like to thank Reviewer 1 for his/her evaluation and comments about our manuscript. We will comment on each of the points presented by the Reviewer, but we must say that a revised version of the manuscript (with track changes) will be submitted after we receive the evaluation and comments from Reviewer 2.

1. Reviewer: I found the paper is difficult to follow because of the language. The authors need to polish the language in this paper.

Answer: The manuscript will be sent to a native speaker for a complete revision

2. In addition, I found most of the ideas in this paper has been discussed in the published paper of de Mahiques et al., 2008, Marine Geology. The authors need to highlight new insights about the sediment sources and ocean circulations.

Answer: We respectfully disagree. The paper of Mahiques et al. (2008) was mainly focused on sediment sources. Apart from a much higher number of samples, the present paper shows an improvement in ocean circulation, far better than the previous work. Nevertheless, we will emphasize these improvements along with the second version.

3. 21 Not true. There is latitudinal trend for Nd isotopes, but not for Pb isotopes

Answer: We respectfully disagree. We performed Correlation analyses for the Pb isotopes, and both $^{207}Pb/^{204}Pb$ vs. Latitude and $^{208}Pb/^{204}Pb$ vs. Latitude provided statistically significant ($p<0.05$) correlations. The Reviewer is correct for $^{206}Pb/^{204}Pb$, which does not present any latitudinal correlation.

40 "Rare Earths" should be "Rare Earth Elements" or "REEs"

Answer: Corrected in the new version

4. 260-264 Pb/Pb? Please correct the ratios in this sentence and the flowing sentences through all manuscript

We acknowledge this comment. When preparing the Word document, we did not realize that all numbers were formatted as hidden. This mistake was corrected in the second version that will be sent after the evaluation by Reviewer 2

5. 347 The sediment samples from Punta del Este basin also have Nd and Pb isotope ratios between Antarctic sediments (a endmember) and Paraná Igneous Province (g endmember). How to exclude this provenances?

Answer: We did not exclude these provenances. On the contrary, we emphasized that sediments from the slope of the Punta del Este basin present an Antarctic source (lines 346-348), and sediments from the shelf have a Río de la Plata signature (lines 351-352)

6. 353-363 The authors try to attibute the Nd and Pb isotope data of Pelotas samples to the influence from Río de la Plate Plume water. But the Pelotas samples display larger range of Nd and Pb isotope compositions than Río de la Plate samples. It is evidence that the Pelotas samples were effected by other sediment provenances. However, the authors did not discuss this point. In addition, the Pelotas sediment samples are obviously located in the area of Brazil Current (Fig. 1), but the authors did not discuss the possible influence from Brazil Current.

Answer: Indeed, the Pelotas basin displays a more extensive range of Nd and Pb isotope values, and that is why we state in the manuscript that they represent a mixture of sediments from the Río de la Plata and Santos Basin. It must be emphasized that the Brazil Current acts only on the outer shelf and upper slope. In this sense, it is not true that all of the Pelotas sediments are located in the area of Brazil Current. All of the samples collected on the inner and middle shelf are outside of the influence of Brazil Current.

One of the main contributions of this paper is that we do not only look at the values of isotopes but also highlight the importance of ocean circulation in

their interpretation. We cannot propose models of transport of sediments flowing against the currents.

7. 363-370 It could not convince me that the Santos samples are related to Precambian metamorphic rocks and granites of the Southeastern Brazilian coast. The Nd and Pb isotope data of the Santos sediments and Precambrian metamorphic rocks as well as granites of the Southeastern Brazilian coast (h, i and j end members) are obviously distinct. The following transport model of the Precambrian Brazilian cannot convince me as well.

Answer: We realized that Figure 7 caused much confusion, and we decided to remove it. The interpretation for a Pre-Cambrian source for the sediments of Santos Basin is based on the extensive work of Mantovanelli et al. (2018), who analyzed two sediment cores at 22.9°S (core 7620) and 25.06°S (core 7616). The authors also summarized data from 21 references of Pre-Cambrian rocks of southern and southeastern Brazil and suspended sediment samples from the Paraíba do Sul River (Roig et al., 2005) (mouth at 21.6°S). Sediments from core 7620 presented $\varepsilon$Nd values (between -17.5 and -16.1) and Sm/Nd ages (between 1.7 and 1.9 Ga) similar to the suspended sediments of the Paraíba do Sul River, which drains mainly Pre-Cambrian rocks. This core shows a signature similar to the metasediments of the Paraíba do Sul Domain.

Concerning core 7616, it presented $\varepsilon$Nd ranging from -11.0 and -9.6 and Sm-Nd age around 1.4 Ga, with similarity with the metasediments from the Coastal Domain. Worth noting that the values obtained in these cores are similar to the samples from Santos Basin that we used in this work. In this sense, we argue that the Pre-Cambrian origin for the sediments of Santos Basin is undoubted.

8. 418-419 It is not true. I observed that the Pelotas samples display mixture characteristics between Santos samples and Punta del Este samples, not Río de la Plate samples.

Answer: As shown in Figure 9 of the original version, the flow in the outer shelf and upper slope between 27°S to 35°S runs to the south. This aspect is due to the thickening of the Brazil Current after receiving intermediate waters from the Santos Bifurcation. In this sense, it is not plausible hydrodynamically that the

Punta del Este sediments can be transported towards the Pelotas Basin. The only northward flow is related to the Plata Plume Water, acting on the inner and middle shelf in that area.

9. 420-421 It is not true. The Santos samples did not have similar Nd and Pb isotope compositions compared to the Precambian rocks (Fig. 7).
Answer: Please refer to our answer in Item 7.

10. What is the Nd and Pb isotope compositions of Two Argentina samples from Brazil-Malvinas confluence (BMC)? Did they inherit the isotope compositions from Malvinas Current or Brazil Current?
Sample 575 is located at 1097 mbsl and presents $\varepsilon$Nd = -2.0, $^{206}$Pb/$^{204}$Pb = 18.480, $^{207}$Pb/$^{204}$Pb = 15.609, and $^{208}$Pb/$^{204}$Pb = 38.366. Its depth indicates that it is under the flow of the Antarctic Intermediate Water (AAIW), carried by the Malvinas Current.
Sample 630 is located at 3620 mbsl and presents $\varepsilon$Nd of -4.2. There are no values for Pb isotopes in that sample. Its depth indicates that it is under the Lower Circumpolar Deep Water (LCDW) or even the Antarctic Bottom Water (AABW). In this sense, it is not carried by the Malvinas Current nor the Brazil Current since it is affected by deep thermohaline circulation.

11. Fig. 1 what is thin red line which is basically parallel with thick red line in this figure?
Answer: The thin red line corresponds to the flow of the Tropical Water. It has been removed in the new version of Figure 1.

References cited
Mantovanelli, S. S., Tassinari, C. C. G., de Mahiques, M. M., Jovane, L., and Bongiolo, E.: Characterization of Nd radiogenic isotope signatures in sediments from the southwestern Atlantic Margin, Frontiers in Earth Science, 6, 74, 10.3389/feart.2018.00074, 2018.

Roig, H. L., Moraes Rego, A. P., Dantas, E. L., Meneses, P. R., Walde, D. H. G., and Goia, S. M. L. C.: Assinatura isotópica Sm-Nd de sedimentos em suspensão: implicações na caracterização da proveniência dos sedimentos do Rio Paraíba do Sul - SP, Revista Brasileira de Geociências, 35, 503-514, 2005.

---

## Author Comment (AC2)

To the Editor of Ocean Science
Ref. 2nd review of the manuscript
"New insights of the influence of ocean circulation on the sedimentary distribution in the Southwestern Atlantic margin (23ºS to 55ºS) based on Nd and Pb isotope fingerprinting"

Dear Sirs,

We acknowledge Reviewer 2 for his/her evaluation and comments about our manuscript. As for Reviewer 1, we will comment on each of the points presented by the Reviewer. In a few weeks, we expect to provide a revised version of the manuscript (with track changes) indicating all of the changes.

1. However, the manuscript is written carelessly.
ANSWER: a full revised manuscript will be provided soon

2. Starting from Line 205, almost all superscripts of Pb and Nd isotopes are missing.
ANSWER: The superscripts were all present in the MS-Word file (.docx). For the reason that we do not know, they were all formatted as "hidden," and we did not notice it until Reviewer 1 complained.

3. First of all, I strongly recommend the authors to show the legends in all Figures. The authors use different marks to represent the data from different areas. In figure 1, there is a legend inside which is good, but in Figure 3 the legend is gone while there still notes in the caption. What's worse happened in figure 7, I can't even find anything in the caption. I really get lost there.
ANSWER: We acknowledge the comment. These mistakes will be corrected in the next version. Concerning Figure 7, we will remove it since it is causing much confusion

4. I have a suggestion which the author can decide to do or not. It is a very long text in section 2 as Morphology, Sedimentary cover, and Ocean Circulation parts. It is good to introduce the basic background and previous work, but I find not all of those parts are very related to this work. My feeling is there is no very

clear focus but just simple descriptions. I suggest the author remove some unrelated parts and move some parts to the discussion section.

ANSWER: We acknowledge the Reviewer. We will try to reduce this topic to the necessary.

5. I also have questions about the methods. In Line 198, the authors state that "Sediment powder (70 mg) was dissolved with HF, HNO3, and HCl acids." However, in Line 211 It then said that "The Nd analyses, here reported as $\varepsilon$Nd, were prepared by standard methods by the analytical procedures described by Sato et al. (1995) and Magdaleno et al. (2017), involving the removal of calcium carbonate, HF–HNO3 dissolution plus HCl cation exchange using a Teflon Powder column to separate REE." These are contradicting each other. Did the author remove the calcium carbonate or not?

ANSWER: We acknowledge the Reviewer. All of the samples were decarbonated (with HCL) before dissolution.  We will include this information in the new version

6. In addition, there a lot of papers reporting reformed Fe-Mn oxides in the sediments near the continent which could be a strong interference to the detrital signals. Did the authors also remove the Fe-Mn oxide coating in their sediment samples? I haven't seen this step in their method.

We did not remove the Fe-Mn coating. The main reason is that as far as we know from all of the previous papers in the area, none of them made the removal of Fe-Mn coating.  The absence of leaching extends to all of the papers on potential sources that we used.  Then, to compare our data with the previous ones, we decided not to remove it.  Also, most of the coating is present in carbonates (foraminifera, for example).  Considering that our analyses were made on carbonate-free samples, we understand that this interference might be reduced, despite not totally eliminated.

7. Besides the chemistry, the authors give the NBS-981 and JNdi results as standard. However, these two standards are used as internal standards to normalize the fractionations. Is there also an external standard to show the analytical reproducibility?

ANSWER: Indeed, the reproducibility analysis was made, using Buffalo River Sediment (NIST-RM8704) (n = 7), with the following results:

$^{143}Nd/^{144}Nd$ = 0.51203 ± 0.00001 (SD)

$^{206}Pb/^{204}Pb$ = 18.846 ± 0.018 (SD)

$^{207}Pb/^{204}Pb$ = 15.646 ± 0.005 (SD)

$^{208}Pb/^{204}Pb$ = 38.503 ± 0.016 (SD)

We added this information to the new version

8. Line 19, "Pb and Nd radiogenic isotopes" should be "radiogenic Pb and Nd isotopes".

ANSWER: Corrected in the new version

Line 33, "Long half-life radiogenic elements, such as Sr, Pb, and Nd" is not a proper description. Not all isotopes of Pb, Nd, and Sr are radiogenic, so you cannot say these elements are radiogenic. Besides that, the long half-life should refer to their radioactive parents, but the daughters.

ANSWER: Corrected in the new version

Once more, we acknowledge both Reviewers for their comments. We hope to provide a fully revised version in few weeks.

Sincerely

Michel M de Mahiques
On behalf of the authors

---

## Author Comment (AC3)

To the Editor of Ocean Science
Ref. 3rd review of the manuscript
"New insights of the influence of ocean circulation on the sedimentary distribution in the Southwestern Atlantic margin (23ºS to 55ºS) based on Nd and Pb isotope fingerprinting"

Dear Sirs,

We acknowledge Reviewer 3 for his/her evaluation and comments about our manuscript. We hope to provide a revised version of the manuscript (with track changes) in a few weeks indicating all of the changes.

1. Broadly, the discussion is poorly written, difficult to follow and possibly need a rewrite.

Answer: We hope to provide a better version shortly

2. The details of study area is just mere description which is not related to this study.

Answer: We acknowledge the comment. Part of the text was removed.

3. Most of the statistical result are not well linked with discussion.

Answer: We acknowledge and will provide a better version

4. Thus, I recommend for a major revision of present version.

Answer: We are working on a fully revised version

5. Authors have claimed "new insight" which they have to specifically highlight in the Abstract. I feel, this is an incremental work based on previous publication by Mahiques et al., 2008, Marine Geology which had no epsilon Nd. It must be highlighted in the paper main text as well.

Answer: We are making changes along with the whole manuscript

6. Introduction need to be more incisive and a proper hypothesis need to be defined. Why there is a need for radiogenic isotope data which will provide a better understanding of the provenance of sediments in the SAM?

Answer: We will provide a better explanation in the reviewed version

7. Study area description can be trimmed down.

Answer:  Please check the answer to topic 2, above

8. I could not see a need for Fig. 2, 3, and 4. How these result helps in assessing the role of ocean circulation on sediment transport? This is completely missing in the discussion.

Answer:  We respectfully disagree. One of the manuscript topics is the utilization of the concept of sediment fingerprinting (Walling, 2013; Miller et al., 2015; Palazon and Navas, 2017).  This concept first involves verifying which variables can be used as fingerprints, demanding a univariate analysis (such as a Mann-Whitney Analysis).  Then, it is necessary to numerically relate the sectors to variables from the fingerprints, which is made with multivariate analysis (such as a Discriminant Analysis).  In this sense, Figure 2 shows a graphic in which it is possible to recognize the variables that present statistically significant differences (all but $^{206}Pb/^{204}Pb$); Figure 3 shows the latitudinal trends of the different variables, and Figure 4 is the graphical expression of the Discriminant Analysis. In this sense, we understand that there is no way to explain the results of the Fingerprinting by removing these figures.

9. I have no clue why authors have not put isotopic ratios with corresponding isotopes e.g. 206Pb/204Pb, 207Pb/204Pb and so on. The same is missing in material and methods.

Answer: As explained previously to the other reviewers, for the reason that we do not know, most of the superscripts were formatted as "hidden" in the MS-Word template. Then, when exporting to pdf, these numbers disappeared. We only noticed this when Reviewer 1 complained about it.

10. Rest of my comments are similar to those raised by other 2 reviewers and suggest authors to consider it carefully, particularly those raised by Rev #1.

Answer:  We hope to provide a better version in a couple of weeks

Once more, we acknowledge both Reviewers for their comments. We hope to provide a fully revised version in few weeks.

Sincerely

Michel M de Mahiques
On behalf of the authors

References cited above

Miller, J. R., Mackin, G., and Orbock Miller, S. M.: Geochemical Fingerprinting, in: Application of Geochemical Tracers to Fluvial Sediment, edited by: Miller, J. R., Mackin, G., and Miller, S. M. O., SpringerBriefs in Earth Sciences, Springer, Cham, 11-51, 2015.

Palazon, L., and Navas, A.: Variability in source sediment contributions by applying different statistic test for a Pyrenean catchment, J Environ Manage, 194, 42-53, 10.1016/j.jenvman.2016.07.058, 2017.

Walling, D. E.: The evolution of sediment source fingerprinting investigations in fluvial systems, Journal of Soils and Sediments, 13, 1658-1675, 10.1007/s11368-013-0767-2, 2013.

---

## Referee Report (RR1)

I found the quality of the manuscript is highly improved. However, the authors need to prepare for the manuscript carefully. There are still lots of mistakes needing to revise.

For example, in introduction:

44 the second biggest river basin in South America?

44 please correct the "m3 s -1"

50 what is "BC"? you did not introduce it here or above.

59-61 are you sure that this paragraph contains only one sentence?

84 did you mention the full name of "SB" in the above sentences?

The authors should double check the whole manuscript.

398-399 they have lower $^{207}Pb/^{206}Pb$ and higher $^{208}Pb/^{206}Pb$ ratios than those located in shallower areas.

I recommend to accept this version of manuscript after minor revision.

---

## Author Response (AR2)

To the Editor of Ocean Science
Ref. Response to the Reviewers
"Control of oceanic circulation on sediment distribution in the Southwestern Atlantic margin (23ºS to 55ºS)"  (OS-2021-44)

Dear Sirs,

Once more, we acknowledge the Topic Editor and reviewers for the positive evaluation of our work.

Concerning the corrections pointed by the Reviewer, we inform that all of the minor changes were corrected.  Also, we found some typo errors and changed the structure of few sentences.   We are providing a tracked-changes version and a consolidated one.

Here is a list of points marked by the reviewer:

**44 the second biggest river basin in South America?**

Word "largest" added in line 37

**44 please correct the "m3 s -1"**

Corrected

**50 what is "BC"? you did not introduce it here or above.**

"Brazil Current (BC)" added in line 43

**59-61 are you sure that this paragraph contains only one sentence?**

Indeed the paragraph was too long. We divided it

**84 did you mention the full name of "SB" in the above sentences?**

Full name and abbreviation provided in line 43

**The authors should double check the whole manuscript.**

Manuscript double-checked. We found some typo errors and rephrased some sentences

**398-399 they have lower 207Pb/206Pb and higher 208Pb/206Pb ratios than those located in shallower areas.**

We changed the sentence

We expect that the new version provided is suitable for publication.

Respectfully

Michel Michaelovitch de Mahiques
     Corresponding author